# COMPRESSED PREDICTIVE INFORMATION CODING

## ABSTRACT

Unsupervised learning plays an important role in many fields, such as machine learning, data compression, and neuroscience. Compared to static data, methods for extracting low-dimensional structure for dynamic data are lagging. We developed a novel information-theoretic framework, Compressed Predictive Information Coding (CPIC), to extract predictive latent representations from dynamic data. Predictive information quantifies the ability to predict the future of a time series from its past. CPIC selectively projects the past (input) into a low dimensional space that is predictive about the compressed data projected from the future (output). The key insight of our framework is to learn representations by balancing the minimization of compression complexity with maximization of the predictive information in the latent space. We derive tractable variational bounds of the CPIC loss by leveraging bounds on mutual information. The CPIC loss induces the latent space to capture information that is maximally predictive of the future of the data from the past. We demonstrate that introducing stochasticity in the encoder and maximizing the predictive information in latent space contributes to learning more robust latent representations. Furthermore, our variational approaches perform better in mutual information estimation compared with estimates under the Gaussian assumption commonly used. We show numerically in synthetic data that CPIC can recover dynamical systems embedded in noisy observation data with low signal-to-noise ratio. Finally, we demonstrate that CPIC extracts features more predictive of forecasting exogenous variables as well as auto-forecasting in various real datasets compared with other state-of-the-art representation learning models. Together, these results indicate that CPIC will be broadly useful for extracting low-dimensional dynamic structure from high-dimensional, noisy time-series data.

## 1 INTRODUCTION

Unsupervised methods play an important role in learning representations that provide insight into data and exploit unlabeled data to improve performance in downstream tasks in diverse application areas Bengio et al. (2013); Chen et al. (2020); Grill et al. (2020); Devlin et al. (2018); Brown et al. (2020); Baevski et al. (2020); Wang et al. (2020). Prior work on unsupervised representation learning can be broadly categorized into generative models such as variational autoencoders(VAEs) (Kingma & Welling, 2013) and generative adversarial networks (GAN) (Goodfellow et al., 2014), discriminative models such as dynamical components analysis (DCA) (Clark et al., 2019), contrastive predictive coding (CPC) (Oord et al., 2018), and deep autoencoding predictive components (DAPC) (Bai et al., 2020). Generative models focus on capturing the joint distribution between representations and inputs, but are usually computationally expensive. On the other hand, discriminative models emphasize capturing the dependence of data structure in the low-dimensional latent space, and are therefore easier to scale to large datasets.

In the case of time series, some representation learning models take advantage of an estimate of mutual information between encoded past (input) and the future (output) (Creutzig & Sprekeler, 2008; Creutzig et al., 2009; Oord et al., 2018). Although previous models utilizing mutual information extract low-dimensional representations, they tend to be sensitive to noise in the observational space. DCA directly makes use of the mutual information between the past and the future (i.e., the predictive information (Bialek et al., 2001)) in a latent representational space that is a linear embedding of the observation data. However, DCA operates under Gaussian assumptions for mutual information

estimation. We propose a novel representation learning framework which is not only robust to noise in the observation space but also alleviates the Gaussian assumption and is thus more flexible.

We formalize our problem in terms of data generated from a stationary dynamical system and propose an information-theoretic objective function for Compressed Predictive Information Coding (CPIC). Instead of leveraging the information bottleneck (IB) objective directly as in Creutzig & Sprekeler (2008) and Creutzig et al. (2009), where the past latent representation is directly used to predict future observations, we predict the compressed future observations filtered by the encoder. It is because that in the time series setting, future observations are noisy, and treating them as labels is not insightful. Specifically, our target is to extract latent representation which can better predict future underlying dynamics. Since the compressed future observations are assumed to only retain the underlying dynamics, better compression thus contributes to extracting better dynamical representation. In addition, inspired by Clark et al. (2019) and Bai et al. (2020), we extend the prediction from single input to a window of inputs to handle high order predictive information.

Moreover, instead of directly estimating the objective information with Gaussian assumption (Creutzig & Sprekeler, 2008; Creutzig et al., 2009; Clark et al., 2019; Bai et al., 2020), we developed variational bounds and a tractable end-to-end training framework based on the neural estimator of mutual information studied in Poole et al. (2019). Note that our inference first leverages the variational boundary technique for self-supervised learning on the time series data. Since it alleviates the Gaussian assumption, it is applicable to a much larger class of dynamical systems.

In CPIC, we also demonstrate that introducing stochasticity into either a linear or nonlinear encoder robustly contributes to numerically better representations in different tasks. In particular, we illustrate that CPIC can recover trajectories of a chaotic dynamical system embedded in high-dimensional noisy observations with low signal-to-noise ratios in synthetic data. Furthermore, we conduct numerical experiments on four real-world datasets with different goals. In two neuroscience datasets, monkey motor cortex (M1) and rat dorsal hippocampus (HC), compared with the state-of-the-art methods, we show that the latent representations extracted from CPIC have better forecasting accuracy for the exogenous variables of the monkey's future hand position for M1, and for the rat's future position for HC. In two other real datasets, historical hourly weather temperature data (TEMP) and motion sensor data (MS), we show that latent representations extracted by CPIC have better forecasting accuracy of the future of those time series than other methods. In summary, the primary contributions of our paper are as follows:

- We developed a novel information-theoretic self-supervised learning framework, Compressed Predictive Information Coding (CPIC), which extracts low-dimensional latent representation from time series. CPIC maximizes the predictive information in the latent space while minimizing the compression complexity.

- We introduced the stochastic encoder structure where we encode inputs into stochastic representations to handle uncertainty and contribute to better representations.

- Based on prior works, we derived the variational bounds of the CPIC's objective function and a tractable, end-to-end training procedure. Since our inference alleviates the Gaussian assumption common to other methods, it is applicable to a much larger class of dynamical systems. Moreover, to the best of our knowledge, our inference is the first to leverage the variational boundary technique for self-supervised learning on time series data.

- We demonstrated that, compared with the other unsupervised based methods, CPIC more robustly recovers latent dynamics in dynamical system with low signal-to-noise ratio in synthetic experiments, and extracts more predictive features for downstream tasks in various real datasets.

## 2 RELATED WORK

Mutual information (MI) plays an important role in estimating the relationship between pairs of variables. It is a reparameterization-invariant measure of dependency:

$$I(X, Y) = \mathbb{E}_{p(x,y)} \left[ \log \frac{p(x|y)}{p(x)} \right] \tag{1}$$

It is used in computational neuroscience (Dimitrov et al., 2011), visual representation learning (Chen et al., 2020), natural language processing (Oord et al., 2018) and bioinformatics (Lachmann et al., 2016). In representation learning, the mutual information between inputs and representations is used to quantify the quality of the representation and is also closely related to reconstruction error in generative models (Kingma & Welling, 2013; Makhzani et al., 2015). Estimating mutual information is computationally and statistically challenging except in two cases: discrete data, as in Tishby et al. (2000) and Gaussian data, as in Chechik et al. (2005). However, these assumptions both severely constrain the class of learnable models (Alemi et al., 2016). Recent works leverage deep learning models to obtain both differentiable and scalable MI estimation (Belghazi et al., 2018; Nguyen et al., 2010; Oord et al., 2018; Alemi et al., 2016; Poole et al., 2019; Cheng et al., 2020).

In terms of representation learning in time series, Wiskott & Sejnowski (2002); Turner & Sahani (2007) targeted slowly varying features, Creutzig & Sprekeler (2008) utilized the information bottleneck (IB) method (Tishby et al., 2000) and developed an information-theoretic objective function. Creutzig et al. (2009) proposed an alternative objective function based on a specific state-space model. Recently, Oord et al. (2018) proposed CPC to extract dynamic information based on an autoregressive model on representations and contrastive loss on predictions. Clark et al. (2019); Bai et al. (2020) proposed unsupervised learning approach to extract low-dimensional representation with maximal predictive information(PI). All of the above unsupervised representation learning models, except for CPC, assume the data to be Gaussian, which may be not realistic, especially when applied to neuroscience datasets (O'Doherty et al., 2017; Glaser et al., 2020), given the non-Gaussianity of neuronal activity. Here, we leverage recently introduced neural estimation of mutual information to construct upper bounds of the CPIC objective and develop an end-to-end training procedure. CPIC enables generalization beyond the Gaussian case and autoregressive models.

Recently, deep encoder networks are leveraged to model nonlinear relations between latent representations and observed data in time series (Chen et al., 2020; Bai et al., 2020; He et al., 2020). However, use of complicated nonlinear encoders induced hinders computational efficiency (Wang et al., 2019). CPIC proposes an efficient representation learning framework for time series that encodes data with maximal predictive information. We also note that there exists several works on the time series modeling from generative modeling perspective. Initially, Fabius & Van Amersfoort (2014) leveraged the recurrent neural network with variational autoencoder to model time series data. Frigola et al. (2014) proposed variational Gaussian-process state-space model. Meng et al. (2021) proposed variational structured Gaussian-process regression network which can efficiently handle more complicated relationships in time series. Most generative modeling inference would depend on the length of time series, while the inference of CPIC depends on the window size $T$, which is more scalable for long time series.

## 3   Compressed Predictive Information Coding

The main intuition behind Compressed Predictive Information Coding (CPIC) is to extract low dimensional representations with minimal compression complexity and maximal dynamical structure. Specifically, CPIC first discards low-level information that is not relevant for dynamic prediction and noise that is more local by minimizing compression complexity (i.e., mutual information) between inputs and representations to improve model generalization. Second, CPIC maximizes the predictive information in the latent space of compressed representations.

Compared with Clark et al. (2019); Bai et al. (2020), CPIC first utilizes stochastic encoder to handle uncertainty of representations, which contributes to more robust representations, and also relieves the Gaussian assumption by constructing bounds of mutual information based on neural estimations. In more detail, instead of employing a deterministic linear mapping function as the encoder to compress data as in Clark et al. (2019), CPIC takes advantage of a stochastic linear or nonlinear mapping function. Given inputs, the stochastic representation follows Gaussian distributions, with means and variances encoded from any neural network structure. A nonlinear CPIC utilizes a stochastic nonlinear encoder which is composed of a nonlinear mean encoder and a linear variance encoder, while a linear CPIC utilizes a stochastic linear encoder which is composed of a linear mean encoder and a linear variance encoder. Note that stochastic representations conditioned on inputs are parameterized as a conditional Gaussian distribution, but the marginal distribution of the representation is a mixture of Gaussian distribution, which is widely recognized as universal approximator of densities.

On the other hand, avoiding the Gaussian assumption on mutual information (Creutzig & Sprekeler, 2008; Creutzig et al., 2009; Clark et al., 2019; Bai et al., 2020), CPIC leverages neural estimations of mutual information. Specifically, we propose differentiable and scalable bounds of the CPIC objective via variational inference, which enables end-to-end training.

Formally, let $X = \{x_t\}, x_t \in \mathbb{R}^N$ be a stationary, discrete time series, and let $X_{\text{past}} = (x_{-T+1}, \ldots, x_0)$ and $X_{\text{future}} = (x_1, \ldots, x_T)$ denote consecutive past and future windows of length T. Then both past and future data are compressed into past and future representations denoted as $Y_{\text{past}} = (y_{-T+1}, \ldots, y_0)$ and $Y_{\text{future}} = (y_1, \ldots, y_T)$ with embedding dimension size $Q$. Similar to the information bottleneck (IB) (Tishby et al., 2000), the CPIC objective contains a trade-off between two factors. The first seeks to minimize the compression complexity and the second to maximize the predictive information in the latent (representation) space. Note that when the encoder is deterministic the compression complexity is deprecated and when the encoder is stochastic the complexity is measured by the mutual information between representations and inputs. In the CPIC objective, the trade-off weight $\beta > 0$ dictates the balance between the compression and predictive information terms:

$$\min_{\psi} \mathcal{L}, \text{ where } \mathcal{L} \equiv \beta(I(X_{\text{past}}; Y_{\text{past}}) + I(X_{\text{future}}; Y_{\text{future}})) - I(Y_{\text{past}}; Y_{\text{future}}) \quad (2)$$

where $\psi$ refer to the model parameters which encode inputs $X$ to latent variables $Y$. Larger $\beta$ promotes a more compact mapping and thus benefits model generalization, while smaller $\beta$ leads to more predictive information in the latent space on training data. This objective function is visualized in Figure 1, where inputs $X$ are encoded into latent space as $Y$ via tractable encoders and the dynamics of $Y$ are learned in a model-free manner.

The encoder $p(Y|X)$ could be implemented by fitting deep neural networks (Alemi et al., 2016) to encode data $X$. Instead, CPIC takes an approach similar to VAEs (Kingma & Welling, 2013), in that it encodes data into stochastic representations. In particular, CPIC employs a stochastic encoder ($g_{\text{enc}}$ in Figure 1) to compress input $x_t$ into $y_t$ as

$$y_t|x_t \sim \mathcal{N}(\mu_t, \text{diag}(\sigma_t^2)), \quad (3)$$

for each time stamp $t$. The mean of $y_t$ is given by $\mu_t = g_{\mu}^{\text{Encoder}}(x_t)$, whereas the variance arises from $\sigma_t = g_{\sigma}^{\text{Encoder}}(x_t)$.

Encoders $g_{\mu}^{\text{Encoder}}$ and $g_{\sigma}^{\text{Encoder}}$ can be any non-linear mapping and is usually modeled using neural network architectures. We use a two-layer perceptron with ReLU activation function (Agarap, 2018) for a nonlinear mapping. In terms of a linear CPIC, we specify the mean of representation as $\mu_t = u^T x_t$. In both linear and nonlinear CPIC setting, if $\sigma_t = 0$, the stochastic encoder reduces to a deterministic encoder.

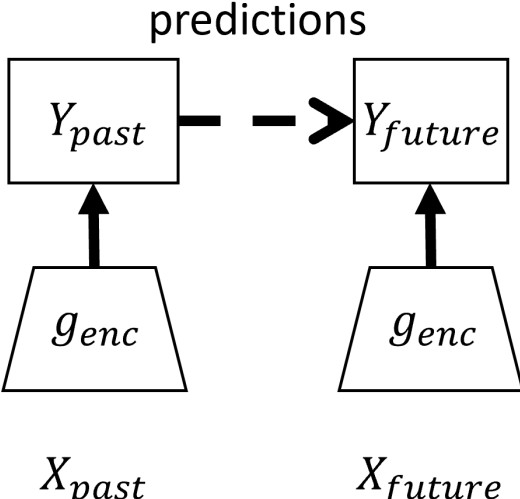

Figure 1: The overall framework of compressed predictive information coding. The encoder compress information of the input $X$ into $Y$ such that the predictive information between $Y_{\text{past}}$ and $Y_{\text{future}}$ is maximized while minimizing the mutual information between $X$ and $Y$.

We extend single input to multiple inputs in the CPIC framework in terms of a specified window size $T$. The selection of window size is discussed in Appendix A. Due to the stationary assumption, the relation between past/future blocks of input data $X(-T), X(T) \in \mathcal{R}^{N \times T}$ and encoded data $Y(-T), Y(T) \in \mathcal{R}^{Q \times T}$ are equivalent, $p_{X(-T),Y(-T)} = p_{X(T),Y(T)}$. Note that $-T$ and $T$ indexes to past and future $T$ data. Without loss of generality, the compression relation can be expressed as $Y(T) = g_{\mu}^{\text{Encoder}}(X(T)) + \xi(T)$, where $\xi(T) \in \mathcal{N}(0, \text{blockdiag}(\text{diag}(\sigma_1^2), \ldots, \text{diag}(\sigma_T^2))$ and noise standard deviation $\sigma_t = g_{\sigma}^{\text{Encoder}}(x_t)$.

# 4 VARIATIONAL BOUNDS OF COMPRESSED PREDICTIVE INFORMATION CODING

In CPIC, since data $X$ are stationary, the mutual information between the input data and the compressed data for the past is equivalent to that for the future $I(X(-T); Y(-T)) = I(X(T); Y(T))$. Therefore, the objective of CPIC can be rewritten as

$$\min \mathcal{L} = \beta I(X(T); Y(T)) - I(Y(-T); Y(T)).$$ (4)

We developed the variational upper bounds on mutual information for the compression complexity $I(X(T); Y(T))$ and lower bounds on mutual information for the predictive information $I(Y(-T); Y(T))$.

## 4.1 UPPER BOUNDS OF COMPRESSION COMPLEXITY

In the section, we derived a tractable variational upper bound (VUB) depending on a single sample and a leave-one-out upper bound (L1Out) (Poole et al., 2019) depending on multiple samples.

**Theorem 1** *By introducing a variational approximation $r(y(T))$ to the marginal distribution $p(y(T))$, a tractable variational upper bound of mutual information $I(X(T); Y(T))$ is derived as $I_{VUB}(X(T); Y(T)) = \mathbb{E}_{X(T)}\big[KL(p(y(T)|x(T)), r(y(T)))\big]$.*

**Theorem 2** *By utilizing a Monte Carlo approximation for variational distribution $r(y(T))$, the L1Out upper bound of mutual information $I(X(T); Y(T))$ is derived as $I_{L1Out}(X(T); Y(T)) = \mathbb{E}\left[\frac{1}{S}\sum_{i=1}^{S}\left[\log\frac{p(y(T)_i|x(T)_i)}{\frac{1}{S-1}\sum_{j\neq i}p(y(T)_i|x(T)_j)}\right]\right]$, where $S$ is the sample size.*

The derivation details are in Appendix B and C. In practice, the L1Out bound depends on the sample size $S$ and may suffer from numerical instability. Thus, we would like to choose the sample size $S$ as large as possible. In general scenarios where $p(y(T)|x(T))$ is intractable, Cheng et al. (2020) proposed a variational version of VUB and L1Out by using a neural network to approximate the condition distribution $p(y(T)|x(T))$. Since the conditional distribution $p(y(T)|x(T))$ is parameterized as a known stochastic/deterministic encoder in CPIC, those variational versions are not taken into consideration.

## 4.2 LOWER BOUNDS OF PREDICTIVE INFORMATION

For the predictive information (PI), we derived lower bounds of $I(Y(-T); Y(T))$ using results in Agakov (2004); Alemi et al. (2016); Poole et al. (2019). In particular, we derived tractable unnormalized Barber and Agakov (TUBA) (Barber & Agakov, 2003) lower bounds depending on a single sample and an infoNCE lower bound (Oord et al., 2018) depending on multi samples. All derivation details are discussed in Appendix D, E and F.

**Theorem 3** *We derived a lower bound on predictive information (PI) I(Y(-T); Y(T)) as $I_{VLB}(Y(-T); Y(T)) = H(Y(T)) + \mathbb{E}_{p(y(-T),y(T))}[\log q(y(T)|y(-T))]$, where $q(y(T)|y(-T))$ is a variational conditional distribution.*

However, this lower bound requires a tractable decoder for the conditional distribution $q(y(T)|y(-T))$ (Alemi et al., 2016). Alternatively we derived a TUBA lower bound (Barber & Agakov, 2003) which is free of the parametrization of decoder.

**Theorem 4** *By introducing a differentiable critic function $f(x,y)$ and a baseline function $a(y(T))$ defined in Appendix E, the TUBA lower bound of predictive information is derived as $I_{TUBA}(Y(-T), Y(T)) = \mathbb{E}_{p(y(-T),y(T))}[\tilde{f}(y(-T), y(T))] - \log\left(\mathbb{E}_{p(y(-T))p(y(T))}[e^{\tilde{f}(y(-T),y(T))}]\right)$ where $\tilde{f}(y(-T), y(T)) = f(y(-T), y(T)) - \log(a(y(T)))$.*

Different forms of the baseline function lead to different neural estimators in the literature such as MINE (Belghazi et al., 2018) and NWJ (Nguyen et al., 2010). On the other hand, all TUBA based

estimators have high variance due to the high variance of $f(x,y)$. Oord et al. (2018) proposed a low-variance MI estimator based on noise-contrastive estimation called InfoNCE. Moreover, there exists other differentiable mutual information estimator including SMILE (Song & Ermon, 2019) and Echo noise estimator (Brekelmans et al., 2019).

**Theorem 5** *In the CPIC setting, the InfoNCE lower bound of predictive information is derived as*

$$I_{infoNCE}(Y(-T); Y(T)) = \mathbb{E}\left[\frac{1}{S}\sum_{i=1}^{S}\log\frac{e^{f(y(-T)_i, y(T)_i)}}{\frac{1}{S}\sum_{j=1}^{S}e^{f(y(-T)_i, y(T)_j)}}\right] \tag{5}$$

The expectation is over $S$ independent samples from the joint distribution: $p(y(-T), y(T))$ following Markov Chain rule in Figure 1 such as $p(y((-T), y(T)) = \int p(x(-T), x(T))p(y(-T)|x(-T))p(y(T)|x(T))dx(-T)x(T)$.

### 4.3 VARIATIONAL BOUNDS OF CPIC

We propose two classes of upper bounds of CPIC based on whether the bounds depend on a single sample or multiple samples. According to the uni-sample and multi-sample bounds derived in Section 4.1 and Section 4.2, we name the first class as uni-sample upper bounds, which take the VUB upper bound of mutual information for the complexity of data compression $I(X(T), Y(T))$ and the TUBA as the lower bound of predictive information in equation 14. Thus we have

$$\mathcal{L}_{\text{UNI}} = \beta\text{KL}(p(y(T)|x(T)), r(y(T))) - I_{\text{TUBA}}(Y(-T), Y(T)). \tag{6}$$

Notice that by choosing different baseline functions, the TUBA lower bound would be equivalent to different mutual information estimator such as MINE and NWJ. The second class is named as multi-sample upper bound, which take advantage of the noise-contrastive estimation approach. The multi-sample upper bound is expressed as

$$\mathcal{L}_{\text{MUL}} = \beta I_{\text{L1Out}}(X(T); Y(T)) - I_{\text{infoNCE}}(Y(-T); Y(T)). \tag{7}$$

Two main differences exist between these classes of upper bounds. First, the performance of multi-sample upper bound depend on batch size while uni-sample upper bounds do not, so when computational budgets do not allow large batch size in training, uni-sample upper bounds may be preferred in training. Secondly, multi-sample upper bound has lower variance than uni-sample upper bounds. Thus, they have different strengths and weaknesses depending on the context. We evaluated the performance of those variational bounds of CPIC in terms of the reconstruction performance in synthetic experiments in Appendix G, and find that with sufficiently large batch size, the multi-sample upper bound would outperform most of the uni-sample upper bounds. Thus, without further specification, we choose the multi-sample upper bound as the variational bounds of CPIC objective in this work. Furthermore, we classify the upper bounds into stochastic and deterministic versions by whether we employ a deterministic or stochastic encoder. Notice that when choosing the deterministic encoder, the compression complexity term (first term) in equation 6 and equation 7 are constant.

## 5 NUMERICAL EXPERIMENTS

In this section, we demonstrate the superior performance of CPIC in both synthetic and real data experiments. We first examine the reconstruction performance of CPIC in noisy observations of a dynamical system (the Lorenz Attractor). The results show CPIC better recovers the latent trajectories from noisy high dimensional observations. Moreover, we demonstrate that maximizing the predictive information(PI) in the compressed latent space is more effective than maximizing PI between latent and observation space as in Creutzig & Sprekeler (2008); Creutzig et al. (2009), and also demonstrate the benefits of the stochastic representation over the deterministic representation. Secondly, we demonstrate better predictive performance of the representation evaluated by linear forecasting. The motivation for using linear forecasting models is that good representations contribute to disentangling complex data in a linearly accessible way (Clark et al., 2019). Specifically, we extract latent representations and then conduct forecasting tasks given the inferred representations

on two neuroscience datasets and two other real datasets. The two neuroscience datasets are multi-neuronal recordings from the hippocampus (HC) while rats navigate a maze (Glaser et al., 2020) and multi-neuronal recordings from primary motor cortex (M1) during a reaching task for monkeys (O'Doherty et al., 2017). The two other real datasets are multi-city temperature data (TEMP) from 30 cities over several years (Gene, 2017) and 12 variables from an accelerater, gyroscope, and gravity motion sensor (MS) recording human kinematics (Malekzadeh et al., 2018). The forecasting tasks for the neuroscience data sets is to predict the future of the relevant exogenous variables from the past neural data, while the forecasting task for the other datasets is to predict the future of those time-series from their past. The results illustrate that CPIC has better predictive performance on these forecasting tasks compared with existing methods.

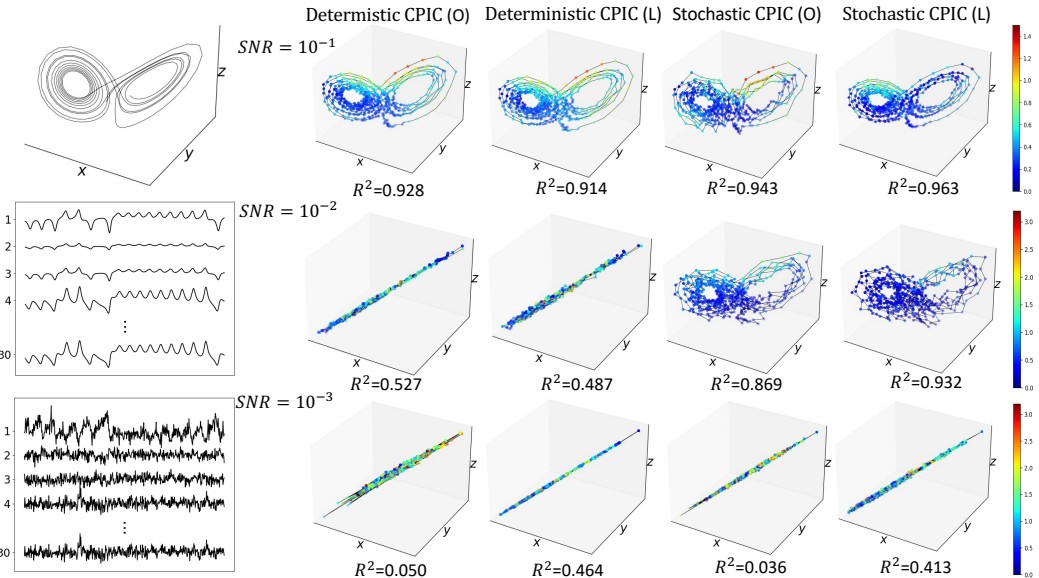

Figure 2: **Left panel.** Top: 3D trajectories of lorenz attractor's ground-truth. Middle: 30D projected trajectory. Bottom: Corrupted 30D trajectory with SNR=0.001. **Right Panel.** 3D trajectories obtained by deterministic CPIC and stochastic CPIC with PI in latent space or between latent and observation space in terms of different SNRs (0.1, 0.008, 0.001). We refer ($L$) to the case with PI in latent space and ($O$) to the case with PI between latent and observation space. We encode the point-wise Euclidean distance between the aligned inferred latent dynamics and the true dynamics into color on trajectories. Color from blue to red corresponds to the distance from short to long respectively. Separate colorbars are used for their corresponding SNRs.

## 5.1 SYNTHETIC EXPERIMENT WITH NOISY LORENZ ATTRACTOR

The Lorenz attractor is a 3D time series that are realizations of the Lorenz dynamical system (Pchelintsev, 2014). It describes a three dimensional flow generated as:

$$\frac{dx}{dt} = \sigma(y - x), \frac{dy}{dt} = f_1(\rho - z) - y, \frac{dz}{dt} = xy - \gamma z \,. \tag{8}$$

Lorenz sets the values $\sigma = 10$, $\rho = 8/3$ and $\gamma = 28$ to exhibit chaotic behavior, as done in recent works (She & Wu, 2020; Clark et al., 2019; Zhao & Park, 2017; Linderman et al., 2017). We simulated the trajectories from the Lorenz dynamical system and show them in the left-top panel in Figure 2. We then mapped the 3D latent signals to 30D lifted observations with a random linear embedding in the left-middle panel and add spatially anisotropic Gaussian noise on the 30D lifted observations in the left-bottom panel. The noises are generated according to different signal-to-noise ratios (SNRs), where SNR is defined by the ratio of the variance of the first principle components of dynamics and noise as in Clark et al. (2019). Specifically, we utilized 10 different SNR levels spaced evenly on a log (base 10) scale between [-3, -1] and corrupt the 30D lifted observations with noise corresponding to different SNR levels. Details of the simulation are available in Appendix G Finally,

we deploy different variants of CPICs to recover the true 3D dynamics from different corrupted 30D lifted observations with different SNR levels, and compare the accuracy of recovering the underlying Lorenz attractor time-series.

We aligned the inferred latent trajectory with the true 3D dynamics with optimal linear mapping due to the reparameterization-invariant measure of latent trajectories. We validated the reconstruction performance based on the $R^2$ regression score of the extracted vs. true trajectories. We first compare the reconstruction performance on different variational bounds of CPIC with the latent dimension size $Q = 3$ and the time window size $T = 4$, and find that multi-sample upper bound outperforms uni-sample upper bounds for almost all of the 10 SNR levels. Thus, we recommend the multi-sample upper bound for CPIC in practice and use that for further results. We also find that, compared to DCA (Clark et al., 2019) and CPC (Oord et al., 2018) CPIC is more robust to noise and thus better extracts the true latent trajectory from the noisy high dimensional observations. The detailed results are reported in Appendix H

In order to demonstrate the benefits of introducing stochasticity in the encoder and maximizing the predictive information in latent space, we considered four variants of CPICs: with stochastic or deterministic encoder, and with predictive information in latent space or between latent and observation space. All four variants of CPIC models utilize the latent dimension size $Q = 3$ and the time window size $T = 4$. For each model and each SNR level, we run 100 replicates with random initializations. We show the aligned latent trajectories inferred from corrupted lifted observation for high, intermediate and low SNR (0.001, 0.01, 0.1) levels of noise with the median $R^2$ scores across 100 replicates in Figure 2. The point-wise distances between the recovered dynamics and the ground-truth dynamics are encoded in the colors from blue to red, corresponding to short to long distance. For high SNR (SNR = 0.1, top-right), all models did a good job of recovering the Lorenz dynamics though the stochastic CPIC with predictive information on latent space had larger $R^2$ than others. For intermediate SNR (SNR = 0.008, middle-right), we see that stochastic CPICs performs much better than the deterministic CPICs. Finally, as the SNR gets lower (SNR = 0.001, bottom-right) all methods perform poorly, but we note that, numerically, considering predictive information in latent space is much better than that between latent and observation space.

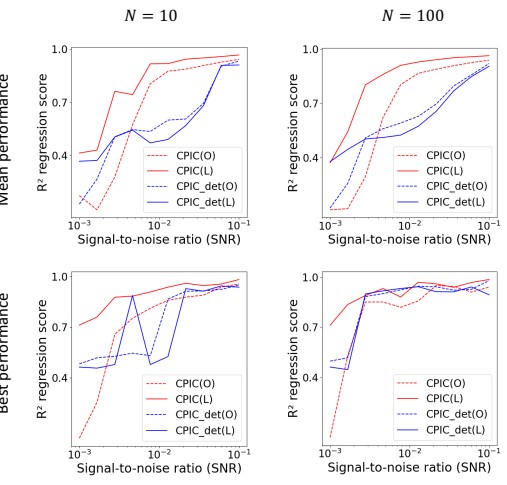

Figure 3: Comparison on $R^2$ scores of latent dynamics regression on 10 SNR levels for four variants of CPIC. The first row shows the mean performance of $R^2$ scores over running N=10/100 different random initializations and the second row shows the best performance over running N=10/100 different random initializations.

To more thoroughly characterize the benefits of stochastic encoding and PI in the latent space, we examined the mean of $R^2$ scores for the four variants on each level of SNR across $N = 10$ and $N = 100$ replicates in the top row of Figure 3. It shows that the CPIC with stochastic representations and PI in latent space robustly outperforms other variants on average. We also report the best $R^2$ scores for the four variants in the sense that we report the $R^2$ score for the model with the smallest training loss across $N$ runs. The bottom row of Figure 3 shows that CPIC with stochastic representation and PI in latent space achieves better reconstruction and robustness to noise than other variants, especially when the number of runs $N$ is small. Even when $N$ is large, stochastic CPIC with PI in latent space greatly outperforms others when the noise level is high. We note that in the case of high-dimensional noisy observations with large numbers of samples common in many modern real-world time series datasets, CPICs robustness to noise and capacity to achieve good results in a small number of runs is a clear advantage. Moveover, we displayed the quantile anaylsis of the $R^2$ scores in Appendix I with consistent result.

## 5.2 REAL EXPERIMENTS WITH DIVERSE FORECASTING TASKS

In this section, we show that latent representations extracted by stochastic CPIC perform better in the downstream forecasting tasks on four real datasets. We compared stochastic CPIC with contrastive predictive coding (CPC) (Oord et al., 2018), PCA, SFA (Wiskott & Sejnowski, 2002), DCA (Clark et al., 2019) and deterministic CPIC. As for CPC, we use a linear encoder for fair comparison. In addition, we compared the result from CPCs and CPICs with nonlinear encoder in which the linear mean encoder is replaced by a multi-layer perceptron. For each model, we extract the latent representations (conditional mean) and conduct prediction tasks on the relevant exogenous variable at a future time step for the neural datasets. For example, for the M1 dataset, we extract a consecutive 3-length window representation of multi-neuronal spiking activity to predict the monkey's arm position in a future time step which is $lag$ time stamps away. The details of experiments are available in Appendix J. Neuroscientists often want to interpret latent representations of data to gain insight into the processes that generate the observed data. Thus, we used linear regression [1] to predict exogenous variables, with the intuition that a simple (i.e., linear) prediction model will only be sensitive to the structure in the data that is easiest to interpret as in (Yu et al., 2008; Pandarinath et al., 2018; Clark et al., 2019). Furthermore, the neuroscience data sets (M1 and HC) present extremely challenging settings for prediction of the exogenous variables due to severe experimental undersampling of neurons due to technical limitations, as well as sizeable noise magnitudes. For these tasks, $R^2$ regression score is used as the evaluation metric to measure the forecasting performance. Four datasets are split into 4:1 train and test data and the forecasting task considered three different lag values (5, 10, and 15). For DCA and deterministic/stochastic CPICs, we took three different window sizes $T = 1, 2, 3$ and report the best $R^2$ scores. Table 1 reports all $R^2$ scores and demonstrates that our stochastic CPIC outperforms all other models except for the case for Temp data with forecasting at lag 15.

Table 1: Comparison between CPC-L (linear encoder), CPC-NL (non-linear encoder), PCA, SFA, DCA, D-CPIC-L (deterministic CPIC with linear encoder), S-CPIC-L (stochastic CPIC with linear encoder), D-CPIC-NL (deterministic CPIC with non-linear encoder), and S-CPIC-NL (stochastic CPIC with non-linear encoder) on $R^2$ regression scores on M1, Hippocampus, Temperature, and Motion sensor datasets with the optimal window size among $T \in [1, 2, 3]$ for three different lag values (5, 10, and 15). $R^2$ regression scores are averaged across five folds.

| Dataset | Lag | CPC-L | CPC-NL | PCA | SFA | DCA | D-CPIC-L | S-CPIC-L | D-CPIC-NL | S-CPIC-NL |
|---------|-----|-------|--------|-----|-----|-----|----------|----------|-----------|-----------|
| M1 | 5 | 0.041 | 0.168 | 0.135 | 0.203 | 0.215 | 0.222 | 0.223 | 0.232 | **0.264** |
|  | 10 | 0.066 | 0.180 | 0.157 | 0.223 | 0.226 | 0.234 | 0.235 | 0.249 | **0.291** |
|  | 15 | 0.068 | 0.152 | 0.145 | 0.199 | 0.200 | 0.202 | 0.203 | 0.226 | **0.252** |
| HC | 5 | 0.025 | 0.018 | 0.007 | 0.112 | 0.113 | 0.120 | 0.127 | 0.145 | **0.150** |
|  | 10 | 0.012 | 0.012 | 0.001 | 0.101 | 0.101 | 0.107 | 0.113 | 0.121 | **0.133** |
|  | 15 | -0.002 | 0.002 | -0.005 | 0.085 | 0.085 | 0.091 | 0.095 | 0.094 | **0.114** |
| Temp | 5 | 0.666 | 0.639 | 0.651 | 0.669 | 0.668 | 0.672 | 0.673 | 0.673 | **0.673** |
|  | 10 | 0.630 | 0.584 | 0.615 | 0.630 | 0.632 | 0.629 | 0.633 | 0.630 | **0.634** |
|  | 15 | **0.624** | 0.529 | 0.581 | 0.623 | 0.622 | 0.620 | 0.621 | 0.621 | 0.621 |
| MS | 5 | 0.281 | 0.184 | 0.107 | -0.051 | 0.443 | 0.247 | 0.457 | 0.290 | **0.483** |
|  | 10 | 0.212 | 0.154 | 0.068 | -0.107 | 0.377 | 0.177 | 0.385 | 0.243 | **0.425** |
|  | 15 | 0.182 | 0.136 | 0.044 | -0.131 | 0.342 | 0.161 | 0.358 | 0.216 | **0.379** |

## 6 CONCLUDING REMARKS

We developed a novel information-theoretic framework, Compressed Predictive Information Coding, to extract representations in sequential data. CPIC balances the maximization of the predictive information in latent space with the minimization of the compression complexity of the latent representation. We leveraged stochastic representations by employing a stochastic encoder and developed variational bounds of the CPIC objective function. We demonstrated that CPIC extracts more accurate low-dimensional latent dynamics and more useful representations that have better forecasting performance in diverse downstream tasks in four real-world datasets. Together, these results indicate that CPIC will yield similar improvements in other real-world scenarios. Moreover, we note that in most real datasets, using nonlinear CPIC would lead to better representation in terms of prediction performance than linear CPIC.

---

[1] https://scikit-learn.org/stable/modules/linear_model.html

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

## APPENDIX

## A    SELECTION OF WINDOW SIZE

Selecting optimal window size $T$ is important for the downstream use of the dynamics. Poor selection of $T$ may cause aliasing artifacts. In general, we nee to select it by cross validation. Furthermore, we can make plots of the predictive information as a function of both window size $T$ and the embedding dimension $Q$ as diagnostic tools.

## B    DERIVATION OF $I_{VUB}$

Directly estimating the compression complexity is intractable, because $I(X(T); Y(T)) := \mathbb{E}_{X(T)}\big[\mathrm{KL}(p(y(T)|x(T)), p(y(T)))\big]$ in which the population distribution $p(y(T))$ is unknown. Thus we introduce a variational approximation to the marginal distribution of encoded inputs

$p(y(T))$, denoted as $r(y(T))$. Due to the non-negativity of the Kullback-Leibler (KL) divergence, the variational upper bound (VUB) is derived as

$$I(X(T); Y(T)) = \mathbb{E}_{X(T)}\big[\text{KL}(p(y(T)|x(T)), r(y(T)))\big] - \text{KL}(p(y(T)), r(y(T)))$$
$$\leq \mathbb{E}_{X(T)}\big[\text{KL}(p(y(T)|x(T)), r(y(T)))\big] = I_{\text{VUB}}(X(T); Y(T)). \quad (9)$$

## C  DERIVATION OF $I_{L1Out}$

Generally, learning $r(y(T))$ was recognised as the distribution density estimation problem (Silverman, 2018), which is challenging. In this setting, the variational distribution $r(y(T))$ is assumed to be learnable, and thus estimating the variational upper bound is tractable. In particular, Alemi et al. (2016) fixed $r(y(T))$ as a standard normal distribution, leading to high-bias in MI estimation. Recently, Poole et al. (2019) utilized a Monte Carlo approximation for variational distribution. In our case, with $S$ sample pairs $(x(T)_i, y(T)_i)_{i=1}^{S}$, $r_i(y(T)) = \frac{1}{S-1}\sum_{j\neq i} p(y(T)|x(T)_j) \approx p(y(T))$ and the L1Out is derived as below:

$$I_{\text{L1Out}}(X(T); Y(T)) = \mathbb{E}\left[\frac{1}{S}\sum_{i=1}^{S}\left[\log\frac{p(y(T)_i|x(T)_i)}{\frac{1}{S-1}\sum_{j\neq i} p(y(T)_i|x(T)_j)}\right]\right]. \quad (10)$$

## D  DERIVATION OF $I_{VLB}$

Similar to Agakov (2004), we replace the intractable conditional distribution $p(y(T)|y(-T))$ with a tractable optimization problem over a variational conditional distribution $q(y(T)|y(-T))$. It yields a lower bound on PI due to the non-negativity of the KL divergence:

$$I(Y(-T); Y(T)) \geq H(Y(T)) + \mathbb{E}_{p(y(-T),y(T))}[\log q(y(T)|y(-T))] \quad (11)$$

where $H(Y)$ is the differential entropy of variable $Y$ and this bound is tight if and only if $q(y(T)|y(-T)) = p(y(T)|y(-T))$, suggesting that the second term in equation 11 equals the negative conditional entropy $-H(Y(T)|Y(-T))$.

However the variational lower bound requires a tractable decoder for the conditional $q(y|x)$. Alternatively, by considering an energy-based variational family for conditional distribution

The conditional expectation in equation 11 can be estimated using Monte Carlo sampling based on the encoded data distribution $p(y(-T), y(T))$. And encoded data are sampled by introducing the augmented data $x(-T)$ and $x(T)$ and marginalizing them out as

$$p(y((-T), y(T)) = \int p(x(-T), x(T))p(y(-T)|x(-T))p(y(T)|x(T))dx(-T)x(T) \quad (12)$$

according to the Markov chain proposed in Figure 1.

## E  DERIVATION OF $I_{TUBA}$

According to Poole et al. (2019), by considering an energy-based variational family to express and conditional distribution $q(y(T)|y(-T))$:

$$q(y(T)|y(-T)) = \frac{p(y(T))e^{f(y(T),y(-T))}}{Z(y(-T))} \quad (13)$$

where $f(x, y)$ is a differentiable critic function, $Z(y(-T)) = \mathbb{E}_{p(y(T))}\big[e^{f(y(T),y(-T))}\big]$ is a partition function, and introducing a baseline function $a(y(T))$, we derived a tractable TUBA lower bound (Barber & Agakov, 2003) of the predictive information as:

$$I(Y(-T), Y(T)) \geq \mathbb{E}_{p(y(-T),y(T))}[\tilde{f}(y(-T), y(T))] - \log\left(\mathbb{E}_{p(y(-T))p(y(T))}[e^{\tilde{f}(y(-T),y(T))}]\right)$$
$$= I_{\text{TUBA}}(Y(-T), Y(T)) \quad (14)$$

where $\tilde{f}(y(-T), y(T)) = f(y(-T), y(T)) - \log(a(y(T)))$ is treated as an updated critic function. Notice that different choices of baseline functions lead to different mutual information estimators.

When $a(y(T)) = 1$, it leads to mutual information neural estimator (MINE) (Belghazi et al., 2018); when $a(y(T)) = Z(y(T))$, it leads to the lower bound proposed in Donsker & Varadhan (1975) (DV) and when $a(y(T)) = e$, it recovers the lower bound in Nguyen et al. (2010) (NWJ) also known as f-GAN (Nowozin et al., 2016) and MINE-f (Belghazi et al., 2018). In general, the critic function $f(x, y)$ and the log baseline function $a(y)$ are usually parameterized by neural networks (Oord et al., 2018; Belghazi et al., 2018): Oord et al. (2018) used a separable critic function $f(x, y) = h_\theta(x)^T g_\theta(y)$, while Belghazi et al. (2018) used a joint critic function $f(x, y) = f_\theta(x, y)$, and Poole et al. (2019) claimed that joint critic function generally performs better than separable critic function but scale poorly with batch size.

## F  DERIVATION OF $I_{infoNCE}$

The derivation of infoNCE in our CPIC setting is trivial by treating $Y(-T)$ and $Y(T)$ as the input and output in the infoNCE formula from the CPC setting (Oord et al., 2018).

## G  DETAILS OF SIMULATION

In this section, we first generated the 3D latent signals according to the Lorenz dynamic system 8 denoted as $X \in \mathbb{R}^{3 \times T}$. We calculated the largest eigenvalue of the covariance matrix of $X$ as dynamic variance denoted as $\sigma_{dynamics}^2$, and the noise variance is $\sigma_{noise}^2 = \sigma_{dynamics}^2 / SNR$ where $SNR$ is signal-to-noise ratio. Then we randomly generate a semi orthogonal matrix $V \in \mathbb{R}^{30 \times 3}$. Then we generated the true 30D signal $VX$ embedded with additive spatially structured white noise, where the noise subspace $V_{noise}$ is generated with median principle angles with respect to dynamics subspaces $V$. The noise covariance is generated via $\Sigma_{noise}$ with the largest eigenvalue $\sigma_{noise}^2$, and then we generate the noisy signal at the $n$th dimension by $[Y_{noisy}]_n \sim \mathcal{N}(v_n^T X, \Sigma_{noise})$, $n = 1, \ldots 30$.

## H  MODEL COMPARISON IN TERMS OF $R^2$ REGRESSION SCORE IN THE NOISY LORENZ ATTRACTOR EXPERIMENT

In this section, the $R^2$ regression scores for CPC, DCA, deterministic & stochastic CPICs (three uni-sample upper bounds in terms of NWJ, MINE, TUBA, and one multi-sample upper bound) for all ten different SNRs are reported in Table 2. It shows that stochastic CPIC with multi-sample upper bound outperforms other approaches in majority of SNRs. It also shows that that CPIC is most robust to the noisy data and thus detect best latent trajectories from noisy observation compared with CPC and DCA.

We also show the aligned latent trajectories inferred from corrupted lifted observation for high, intermediate and low SNR (0.001, 0.01, 0.1) levels of noise with the median $R^2$ scores across 100 replicates for PCA and DCA (as the extension of Figure 2) in Figure 4. The point-wise distances between the recovered dynamics and the ground-truth dynamics are encoded in the colors from blue to red, corresponding to short to long distance. It show that stochastic CPIC outperforms both PCA and DCA.

## I  COMPARISON ON $R^2$ SCORES OF LATENT DYNAMICS REGRESSION FOR NOISY LORENZ ATTRACTOR IN TERMS OF QUANTILE ANALYSIS

We displayed the medium performance (with the inter-quantile range as the error bars) of $R^2$ scores of latent dynamics regression for noisy Lorenz attractor in Figure 5.

## J  DETAILS OF REAL-WORLD EXPERIMENTS

The four real data are Monkey motor cortical dataset (M1), Rat hippocampal data (HC), Temperature dataset (Temp) and Accelerate dataset (MS).

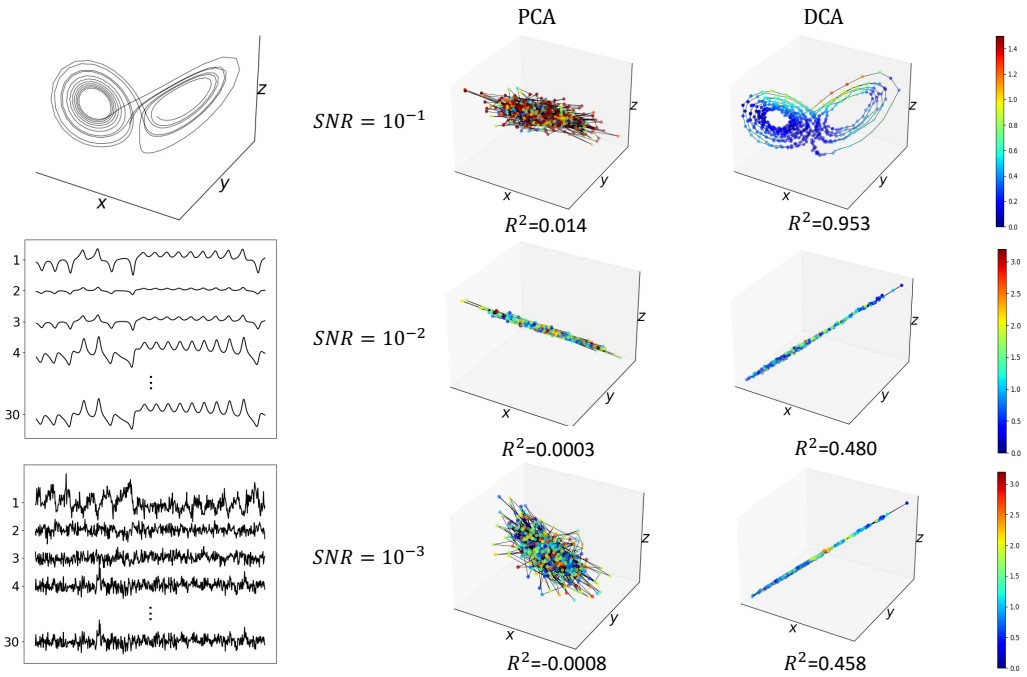

Figure 4: **Left panel.** Top: 3D trajectories of lorenz attractor's ground-truth. Middle: 30D projected trajectory. Bottom: Corrupted 30D trajectory with SNR=0.001. **Right Panel.** 3D trajectories obtained by PCA and DCA in terms of different SNRs (0.1, 0.008, 0.001). We encode the point-wise Euclidean distance between the aligned inferred latent dynamics and the true dynamics into color on trajectories. Color from blue to red corresponds to the distance from short to long respectively. Separate colorbars are used for their corresponding SNRs.

Table 2: $R^2$ regression scores for CPC, DCA, deterministic & stochastic CIPCs including three uni-sample upper bounds (UNI): NWJ, MINE, TUBA, and one multi-sample upper bound (MUL) for all ten different SNRs

| SNR | CPC | DCA | CPIC | | | | | | | |
|---|---|---|---|---|---|---|---|---|---|---|
| | | | Deterministic | | | | Stochastic | | | |
| | | | UNI | | | MUL | UNI | | | MUL |
| | | | NWJ | MINE | TUBA | | NWJ | MINE | TUBA | |
| 0.001 | 0.132 | 0.458 | **0.554** | 0.543 | 0.547 | 0.482 | 0.539 | 0.550 | 0.553 | 0.459 |
| 0.00167 | 0.195 | 0.466 | 0.539 | 0.538 | 0.574 | 0.430 | 0.573 | 0.569 | 0.571 | **0.576** |
| 0.00278 | 0.265 | 0.473 | 0.573 | 0.573 | 0.573 | 0.413 | 0.587 | 0.583 | **0.590** | 0.588 |
| 0.00464 | 0.344 | 0.478 | 0.579 | 0.562 | 0.584 | 0.438 | **0.598** | 0.583 | 0.556 | 0.593 |
| 0.00774 | 0.421 | 0.480 | 0.597 | 0.559 | 0.515 | **0.912** | 0.582 | 0.579 | 0.589 | 0.598 |
| 0.01292 | 0.491 | 0.484 | 0.587 | 0.596 | 0.597 | 0.468 | 0.580 | 0.563 | 0.592 | **0.923** |
| 0.02154 | 0.547 | 0.486 | 0.590 | 0.596 | 0.592 | 0.688 | 0.568 | 0.599 | 0.864 | **0.930** |
| 0.03594 | 0.592 | 0.491 | 0.587 | 0.912 | 0.594 | 0.923 | 0.937 | 0.632 | 0.907 | **0.951** |
| 0.05995 | 0.635 | 0.952 | 0.933 | 0.837 | 0.936 | 0.474 | 0.970 | 0.939 | 0.896 | **0.970** |
| 0.1 | 0.671 | 0.953 | 0.920 | 0.893 | 0.889 | 0.922 | 0.926 | 0.910 | 0.854 | **0.989** |

## J.1 MONKEY MOTOR CORTICAL DATASET

O'Doherty et al. (2017) released multi-electrode spiking data for both M1 and S1 for two monkeys during a continuous grid-based reaching task. We used M1 data from the subject "Indy" (specifically, we used the file "indy_20160627_01.mat"). We discarded single units with fewer than 5,000 spikes, leaving 109 units. We binned the spikes into non-overlapping bins , square-root transformed the data and mean-centered the data using a sliding window 30 s in width.

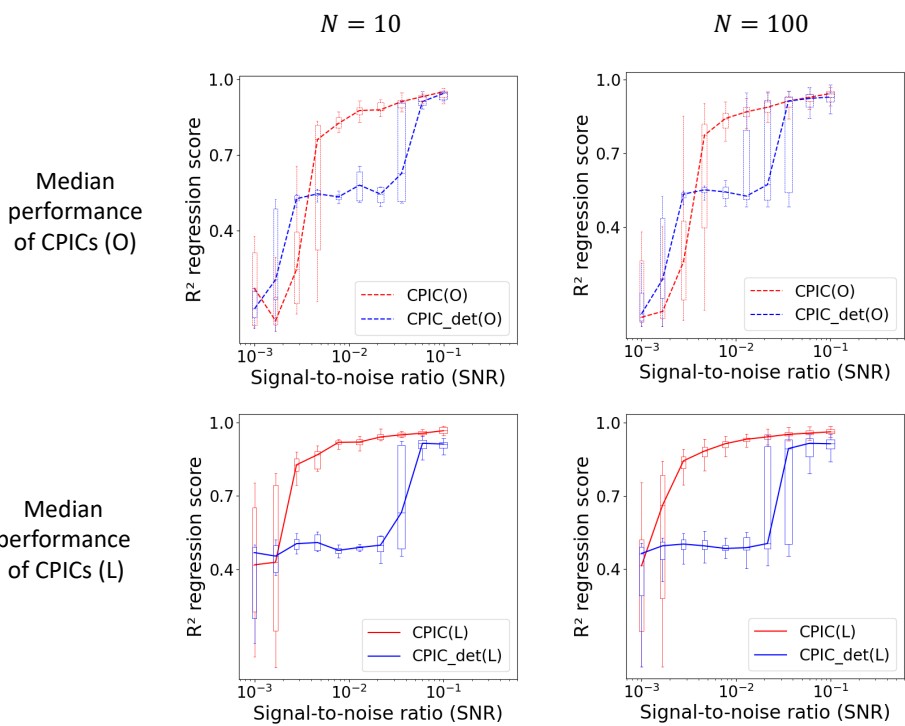

Figure 5: Comparison on $R^2$ scores of latent dynamics regression on 10 SNR levels for four variants of CPIC. The first row shows the medium performance (with the inter-quantile range as the error bars) of $R^2$ scores for CPICs (O) with PI between latent and obersvation space over running N=10/100 different random initializations, and the second row shows the performance for CPICs (L) with PI in latent space.

## J.2 RAT HIPPOCAMPAL DATA

Glaser et al. (2020) released the original data. The data consist of 93 minutes of extracellular recordings from layer CA1 of dorsal hippocampus while a rat chased rewards on a square platform. We discarded single units with fewer than 10 spikes, leaving 55 units. We binned the spikes into non-overlapping 50 ms bins, then square-root transformed the data.

## J.3 TEMPERATURE DATASET

The temperature dataset consists of hourly temperature data for 30 U.S. cities over a period of 7 years from OpenWeatherMap.org. We downsampled the data by a factor of 24 to obtain daily temperatures.

## J.4 ACCELEROMETER DATASET

Malekzadeh et al. (2018) released accelerometer data which records roll, pitch, yaw, gravity x, y, z, rotation x, y, z and acceleration x, y, z for a total of 12 kinematic variables. The sampling rate is 50 Hz. We used the file "sub_19.csv" from "A_DeviceMotion_data.zip".

## J.5 FORECASTING TASK

The forecasting task is the same in Clark et al. (2019). We use the extracted consecutive 3-length window representation of endogenous data to forecast the future relevant exogenous variables at log n. In M1 and HC, the endogenous variables are processed spiking data, and the exogenous variables are location data. In Temp and MS, we assume endogenous variables and exogenous variables are the same, 30 U.S. cities' hourly temperature for Temp data and 12 kinematic variables for MS data.

