# OpenReview forum: "Compressed Predictive Information Coding"
_ICLR.cc/2023/Conference — Submitted to ICLR 2023_

### Official Review · Reviewer_jyD8 · 2022-10-24

**Confidence:** 3
**Correctness:** 3
**Technical Novelty And Significance:** 3
**Empirical Novelty And Significance:** 3
**Recommendation:** 6

**Clarity, Quality, Novelty And Reproducibility:**

In general, this method extends the prior works and propose a variational bound to reduce the computation cost. The paper has both theoretical and empirical results. Most related methods are included for comparison. The code is not provided. The method is not straightforward to re-implement.

**Strength And Weaknesses:**

Pros:
1. The variational bound alleviates the cost of predictive information. It can potentially be used in more scenarios as a plug-in module.
2. Operating in the latent space makes more sense than input space and employs the power of non-linear layers.
3. Various experients and real-world datasets are shown.

Cons:
1. It seems that DAPC has a very similar framework, also with a deterministic and probabilistic encoders (also a VAE?). Could you further elaborate on the differences?
2. Could you further justify the stationarity assumption? Is it commonly seens in time series analysis?
3. When you demonstrate the qualitative results (e.g. synthetic data), maybe you can also show some results from other methods.
4. You can also show some time cost numbers to demonstrate the advantage of your method speed-wise.

**Summary Of The Paper:**

This paper aims at extracting low-dimensional structure for dynamic data, especially for time series data. The proposed method, Compressed Predictive Information Coding (CPIC), both minimizes compression complexity and maximizes the predictive information
in the latent space. This work extends the prior works dynamical components analysis and deep autoencoding predictive components. This work includes the analysis for both linear and non-linear encoding. The core idea is to estimate the latent predictive information without heavy matrix computation, but with cheap approximations. The resulting model achieves good performance on synthetic lorenz attractor, two neuroscience datasets and other real-world scenarios.

**Summary Of The Review:**

This paper improves over previous method, and potentially reduces the time cost. Operating in the latent space also takes advantage of the large capacity brought by deep models. It would be better if the method is also tested on larger and more mainstream time series datasets.

---

> ### Author Response · Authors · 2022-11-16
> **Response to Reviewer jyD8**
>
> Thanks for supporting our paper and providing valuable comments. We've modified our paper based on your comments. Your concerns are addressed as follows.
>
> **Question 1**
> It seems that DAPC has a very similar framework, also with a deterministic and probabilistic encoders (also a VAE?). Could you further elaborate on the differences?
>
> **Answer 1**
> We thank the reviewer asking for the difference between CPIC and DAPC. First of all, DAPC assumes data with Gaussian assumption while CPIC does not. Thus CPIC is more flexible than DAPC. Second, as for the probabilistic DAPC, CPIC is treated as a discriminative model with a deterministic/stochastic encoder while the probabilistic DAPC is treated as a generative model in a VAE framework (i.e., it has both encoder and decoder). Third, as for the probabilistic DAPC, it requires to model the covariance of data within a window size while CPIC does not need to model that. We also briefly discussed it in the last second paragraph of the related work section in the revised manuscript.
>
> **Question 2**
> Could you further justify the stationarity assumption? Is it commonly seen in time series analysis?
>
> **Answer 2**
> We thank the reviewer for raising question on the stationarity assumption. The stationarity assumption is widely used in time-series applications which can significantly simplify our model. To justify the stationarity assumption, the raw data is processed before we model them. The data processing including de-trending and data normalization. The data in our real applications have been processed before we model them, and thus those data are assumed to be stationary.
>
> **Question 3**
> When you demonstrate the qualitative results (e.g. synthetic data), maybe you can also show some results from other methods.
>
> **Answer 3**
> We thank the reviewer for the suggestion on the model comparison. We provided the qualitative results for two other models (PCA and DCA) in Appendix H.
>
> **Question 4**
> You can also show some time cost numbers to demonstrate the advantage of your method speed-wise.
>
> **Answer 4**
> We thank the reviewer for pointing out the running time benefits. Compared with the state-space model such as CPC, our CPIC is more scalable. Due to the stochastic optimization, the inference does not depend on the length of sequence $N$ but depends on the window size $T$, while CPC does depend on $N$. In practice, $T$ is significantly than $N$, thus CPIC is more scalable for long time series. We have discussed it in the last paragraph of related work section. Specifically:
> Most generative modeling inference would depend on the length of time series, while the inference of CPIC depends on the window size T , which is more scalable for long time series."
>
> **Question 5**
> The code is not provided. The method is not straightforward to re-implement.
>
> **Answer 5**
> We thank the reviewer asking for the code. We will provide our code in the github.

---

### Official Review · Reviewer_aQPA · 2022-10-24

**Confidence:** 4
**Correctness:** 3
**Technical Novelty And Significance:** 3
**Empirical Novelty And Significance:** Not applicable
**Recommendation:** 6

**Clarity, Quality, Novelty And Reproducibility:**

The writing of the paper has some drawbacks that make it harder to follow up the ideas. Some examples are in the following:
1) use \citep when citations are inside of the sentence
2) sequential data is not always dynamic and not always timeseries (using i.e. in this case is not justified)
3) some terms are used without introduction, for example "low-level information" and "compression complexity" in section3
4) it would be nice to specify what activation functions are used in the encoders
5) notation on page4, in the end, is not very nice: usually p(X(T), Y(T)) will denote a probability of having this tuple, not the probability distribution, which is meant there
6) typo in paragraph after Theorem3: q(y(T)|y(-T)), not q(y|x)
7) in section4.3 it is an upper bound on CPIC, not lower bound
8) in section4.3 it was unclear why there is a one-sample bound and multiple-samples bound. It should be explained more precisely.
9) in section5.2 it is mentioned that "predictions are conducted by linear regression to emphasize the structure learned by the unsupervised methods". It is a confusing formulation and it should be described in more details what is the reason to use linear regression.

The novelty of the paper is about formulating information bottleneck loss for the time series forecasting task.

The code for the experiments is not provided, reproducibility might be hard.

**Strength And Weaknesses:**

The paper is proposing an information bottleneck type of the loss for a different from usual setup. The time series forecasting is a complicated task that is an active research area and novel approaches are always interesting.

The paper uses existing estimators and bounds on mutual information, thus the main novelty of the paper is the formulation of the information bottleneck loss and finding methods that allow to compute it. The evaluation shows that the proposed method performs better than two existing approaches, though still with not very high performance.

**Summary Of The Paper:**

The paper proposes a way to optimize time series forecasting model through adapted information bottleneck loss. In particular, it is proposed to learn low dimensional representations of the input time series (using variational encoders) via maximizing the mutual information between representations of the input and target and minimizing mutual information between input and representation. The approach is termed compressed predictive information coding (CPIC) and the authors propose two different computable bounds, based on the existing estimators of mutual information, that can be used to train the models. Further, authors explore the empirical performance of the proposed bounds, first on the artificially generated data with clear structure, to check the ability of the representations learned with CPIC to reconstruct data. The second experimental evaluation is done on the real data where forecasting is made using linear regression on the learned representations.

**Summary Of The Review:**

The paper formulates an information bottleneck loss for timeseries forecasting problem, using representations learned with variational encoders. The existing estimators combined to produce a computable bound for the proposed loss and further this bound evaluated for the task of reconstruction and forecasting.
Overall, my main concern is about the reported evaluation results, which seem to be not very high. Since the core of the paper is formulation of the loss, the empirical evaluation should be very convincing.
Moreover, why the reported results for existing methods (CPC and DCA) are so much worse, nearly twice on average or even more? Are these methods so bad for the problems selected for experiments? Or only with the SNR that is very low? The reported results are also very low for the real world data. I wonder what is the state-of-the-art scores on the problems considered.

----
I thank the authors for the clarifications. I will stick with my score after the internal discussion.

---

> ### Author Response · Authors · 2022-11-16
> **Response to Reviewer aQPA (part 1)**
>
> Thank you for supporting our paper and providing valuable comments. We've modified our paper based on your comments. Your concerns are addressed as follows.
>
> **Question 1**
> Use citep when citations are inside of the sentence
>
> **Answer 1**
> We thank the reviewer to provide suggestion on the format of citations. We have followed the suggestions in the revised manuscript.
>
> **Question 2**
> Sequential data is not always dynamic and not always timeseries (using i.e. in this case is not justified)
>
> **Answer 2**
> We thank the reviewer asking for the clarity of sequential data. Since our work mainly focus on time series, we reword sequential data or dynamic data to time series.
>
> **Question 3**
> Some terms are used without introduction, for example "low-level information" and "compression complexity" in section3.
>
> **Answer 3**
> We thank the reviewer for the clarity of some technical terms. We have added explanation on those terms in the revised manuscript. On the bottom of Page 3:
> "Specifically, CPIC first discards low-level information that is not relevant for dynamic prediction and noise that is more local by minimizing compression complexity (i.e., mutual information) between inputs and representations to improve model generalization. "
>
> **Question 4**
> It would be nice to specify what activation functions are used in the encoders
>
> **Answer 4**
> We thank the reviewer for the clarity of the activation functions. We added this information (ReLU activation function [1]) for the encoder in Section 3.
>
> [1] Deep learning using rectified linear units (ReLU), arXiv preprint arXiv:1803.08375, 2018 - arxiv.org
>
> **Question 5**
> Notation on page4, in the end, is not very nice: usually p(X(T), Y(T)) will denote a probability of having this tuple, not the probability distribution, which is meant there.
>
> **Answer 5**
> We thank the reviewer for suggesting this improved clarity of the notation. We have changed p(X(T), Y(T)) to $p_{X(T), Y(T)}$ to express the probability distribution in the revised manuscript.
>
> **Question 6**
> Typo in paragraph after Theorem3: $q(y(T)|y(-T))$, not $q(y|x)$
>
> **Answer 6**
> We thank the reviewer pointing out the typo. We have corrected it in the revised manuscript.
>
> **Question 7**
> In section 4.3 it is an upper bound on CPIC, not lower bound.
>
> **Answer 7**
> We thank the reviewer pointing out the typo. We have corrected it in the revised manuscript.
>
> **Question 8**
> In section 4.3 it was unclear why there is a one-sample bound and multiple-samples bound. It should be explained more precisely.
>
> **Answer 8**
> We thank the reviewer asking for the difference between the one-sample bound and multiple-samples bound. We believe this is addressed by **AR0.3** above. As note in **AR0.3**, we have clarified this issue in Section 4.3 (middle of Page 6):". First, the performance of multi-sample upper bound depend on batch size while uni-sample upper bounds do not, so when computational budgets do not allow large batch size in training, uni-sample upper bounds may be preferred in training. Secondly, multi-sample upper bound has lower variance than uni-sample upper bounds. Thus, they have different strengths and weaknesses depending on the context."
>
>
> **Question 9**
> In section 5.2 it is mentioned that "predictions are conducted by linear regression to emphasize the structure learned by the unsupervised methods". It is a confusing formulation and it should be described in more details what is the reason to use linear regression.
>
> **Answer 9**
> We thank the reviewer raising the lack of clarity on this issue. We believe this is addressed by **AR0.4:** above.
>
> **Question 10**
> The code for the experiments is not provided, reproducibility might be hard.
>
> **Answer 10**
> We thank the reviewer for pointing out that the lack of software would negatively impact reproducibility. We believe this is addressed by **AR0.1:** above.

---

> > ### Comment · Reviewer_aQPA · 2022-11-18
> > **Reply**
> >
> > I thank the authors for a thorough reply and appreciate the revision of the manuscript.
> >
> > I support the opinion that methods should be applied on the real data and it is important to take into account the needs of the domain.

---

> ### Author Response · Authors · 2022-11-16
> **Response to Reviewer aQPA (part 2)**
>
> **Question 11**
> **[a]** Overall, my main concern is about the reported evaluation results, which seem to be
> not very high. Since the core of the paper is formulation of the loss, the empirical evaluation should
> be very convincing. **[b]** Why the reported results for existing methods (CPC and DCA) are so much worse, nearly twice on average or even more? Are these methods so bad for the problems selected for experiments? Or only with the SNR that is very low? The reported results are also very low for the real world data. I wonder what is the state-of-the-art scores on the problems considered.
>
> **Answer 11**
> We thank the reviewer asking the questions on the experiments. We believe that **[a]** is addressed by **AR0.4** above.
>
> **[b]** The reason why the results for CPC and DCA are very pool is because CPC ignores the predictive capability of latent representation which are very important for downstream tasks. As for DCA, although DCA considered predictive capability, it has strong assumption including Gaussian assumption for data and linear assumption for encoding model. Extracting the useful dynamic representations are challenges. The SOTA low dimension reduction model for neuroscience experiments is DCA [4] and the SOTA deep learning based method is CPC. We compared the model performance with both SOTA methods and achieved better performance for both of them.

---

### Official Review · Reviewer_3inW · 2022-10-24

**Confidence:** 3
**Correctness:** 2
**Technical Novelty And Significance:** 2
**Empirical Novelty And Significance:** 2
**Recommendation:** 3

**Clarity, Quality, Novelty And Reproducibility:**

The writing generally needs improvement. For example, there are countless typos, imprecise technical language and strange grammatical structures. However, there are also more major stylistic issues:
 - Much of the Introduction is dedicated to discussing related works in a way that doesn't directly pertain to the authors' proposed method. These parts should be moved into the related works section. The clearest example is the second paragraph, which should be (perhaps even verbatim) moved to Section 2.
 - Similarly, most of the first paragraph of Section 3 should also be moved to the Related Works section.
 - The authors' work seems to be very closely related to variational recurrent auto-encoders and variational state-space models, yet they are not discussed in the related works. Could the authors please comment on this?
 - In Section 3, the authors write: "A nonlinear CPIC refers to a stochastic nonlinear encoder including a nonlinear mean encoder and a linear variance encoder, while a linear CPIC refers to a stochastic linear encoder in which it replaces the linear mean encoder by a nonlinear mean encoder." - I highlight this particular sentence because it is supposed to describe the variants of CPIC that are later presented, but is very challenging to parse and probably contains errors.
 - In Thm 4, the definitions of "critic function" and "baseline function" are missing.
 - The authors only present results for the multi-sample upper bounds in the main text. Hence, I don't think presenting the univariate bounds is useful, and their discussion should be moved to the appendix.
 - Probably every section title in Section 5 should be reworded. In particular, "Numerical demonstration of the superiority of CPIC" should be renamed to "Results" or "Experiments".
 - "The motivation for using linear forecasting models is that good representations contribute to disentangling complex data in a linearly accessible way" - what does "linearly accessible way" mean?
 - Just below Eq 3: "$U \in \mathcal{R}^{N \times D}$" - what is $U$?
 - What is $f_1$ and $\gamma$ in Eq 8? In the sentence below, what is $\beta$?
 - Label font sizes in Figure 2 and the legend font sizes in Figure 3 should be increased as they are currently hard to read.
 - Instead of showing mean and best performance, Figure 3 should show median performance with the inter-quartile range as the error bars to give a better idea of performance.
 - "Finally, as the SNR gets lower (SNR = 0.001, bottom-right) all methods perform poorly, but we note that, numerically, considering predictive information in latent space is much better than that between latent and observation space." - why is stochasticity better numerically? There is no justification given for this claim.


**Strength And Weaknesses:**

## Strengths
The proposed approach is well-motivated and an interesting idea. It is quite simple; hence, I wonder if it has already been proposed elsewhere. However, I am not familiar enough with the sequential prediction literature to be able to comment on the originality of the approach though.

Some of the empirical results also seem promising.

## Weaknesses
In addition to the list below, the paper suffers from severe stylistic and clarity issues; see the section below.
 - In Sections 4.1 & 4.2, the authors state four "theorems" that give certain variational bounds on the intractable terms in their proposed loss function in Eq 4. However, it is not entirely clear what the authors are claiming here as their original contribution. The statements in Thm 1 and 3 follow trivially using elementary information-theoretic arguments in two lines (Eqs 9 and 11 in the appendix). The statements of Thm 2 and 4, and 5 are essentially taken from other works with trivial substitutions. Could the authors please comment on what exactly they are claiming as their contribution here?
 - The authors never formally state the exact models used for their experiments, making their results impossible to interpret.

**Summary Of The Paper:**

The authors propose a method for self-supervised time-series prediction called Compressed Predictive Information Coding (CPIC). The core idea is that a sequence of data points are mapped to a Gaussian latent space via some encoder function. Then, these latent representations can be used for downstream tasks, e.g. regression. The authors propose minimizing the difference between two computationally intractable mutual information terms to train the model. To make their approach practically applicable, the authors propose to minimize a variational upper bound to the original objective. The authors test their approach on synthetic and some real-world regression problems.

**Summary Of The Review:**

The authors present a nice, well-motivated idea. However, it is unclear what theoretical contributions they claim, the description of their empirical methodology is incomplete, and the presentation quality is generally low.

---

> ### Author Response · Authors · 2022-11-16
> **Response to Reviewer 3inW (part 1)**
>
> Thanks for providing valuable comments and suggestions on the clarity of our work. We've modified our paper based on your comments. Your concerns are addressed as follows.
>
> **Question 1**
> In Sections 4.1 \& 4.2, the authors state four "theorems" that give certain variational bounds on the intractable terms in their proposed loss function in Eq 4. However, it is not entirely clear what the authors are claiming here as their original contribution. The statements in Thm 1 and 3 follow trivially using elementary information-theoretic arguments in two lines (Eqs 9 and 11 in the appendix). The statements of Thm 2 and 4, and 5 are essentially taken from other works with trivial substitutions. Could the authors please comment on what exactly they are claiming as their contribution here?
>
> **Answer 1**
> We thank the reviewer for pointing out the lack of clarity on this issue. We believe this issue is addressed by above by **AR0.2**. We note that the goal of this work is to develop a novel information-theoretic self-supervised learning framework for representation learning with CPIC objective. Estimating the differential mutual information is not the key point.
>
> **Question 2**
> The authors never formally state the exact models used for their experiments, making their results impossible to interpret.
>
> **Answer 2**
> We thank the reviewer for pointing out the lack of clarity on the model information in our experiments. We believe this issue is largely addressed by **AR0.1** above. The CPIC model is described in Section 3 and its objectives are (6) and (7) corresponding to uni-sample or multi-sample upper bounds. In the synthetic experiment, we stated that we first compared the CPIC with uni-sample and multi-sample upper bounds, DCA (Clark et al 2019) and CPC (Oord et al 2018) in Appendix H. We also conducted an ablation study for different variant of CPICs in terms of using deterministic or stochastic encoder, leveraging the predictive information (PI) in latent space or the PI between latent and observation space shown in Figure 2 and 3. Finally, we stated that we compared our CPICs with CPCs (Oord et al 2018), PCA, SFA (Wiskott \& Sejnowski 2002), DCA (Clark et al 2019) in real experiments with results in Table 1. The citations for those methods provide relevant details.
>
> **Question 3**
> Much of the Introduction is dedicated to discussing related works in a way that doesn't directly pertain to the authors' proposed method. These parts should be moved into the related works section. The clearest example is the second paragraph, which should be (perhaps even verbatim) moved to Section 2.
>
> **Answer 3**
> We thank the reviewer for providing suggestions on the organization of this paper. As the reviewer suggested, we modified the introduction and moved some of works from introduction to related work.
>
> **Question 4**
> Similarly, most of the first paragraph of Section 3 should also be moved to the Related Works section.
>
> **Answer 4**
> We thank the reviewer for the suggestion on the organization of this paper. We have put some content in Section 3 to Section 2.
>
> **Question 5**
> The authors' work seems to be very closely related to variational recurrent auto-encoders and variational state-space models, yet they are not discussed in the related works. Could the authors please comment on this?
>
> **Answer 5**
> We thank the reviewer asking for the relation between our CPIC and variational recurrent auto-encoders/variational state-space models. Although our work focus on the discriminant models, we added the discussion for those generative models (variational recurrent auto-encoders/variational state-space models) in the Section 2 on Page 3.
>
> **Question 6**
> In Section 3, the authors write: "A nonlinear CPIC refers to a stochastic nonlinear encoder including a nonlinear mean encoder and a linear variance encoder, while a linear CPIC refers to a stochastic linear encoder in which it replaces the linear mean encoder by a nonlinear mean encoder." - I highlight this particular sentence because it is supposed to describe the variants of CPIC that are later presented but is very challenging to parse and probably contains errors.
>
> **Answer 6**
> We thank the reviewer for asking for the clarity of the two variants of CPIC. We rephrased it in the revised manuscript. Specifically, on Page 3: "A nonlinear CPIC utilizes a stochastic non- linear encoder which is composed of a nonlinear mean encoder and a linear variance encoder, while a linear CPIC utilizes a stochastic linear encoder which is composed of a linear mean encoder and a linear variance encoder."

---

> ### Author Response · Authors · 2022-11-16
> **Response to Reviewer 3inW (part 2)**
>
> **Question 7**
> In Thm 4, the definitions of "critic function" and "baseline function" are missing.
>
> **Answer 7**
> We thank the reviewer for pointing out the lack of definitions. Due to the page limit, the definition of both functions can be found in Appendix E. Moreover, we discussed several critic functions and baseline functions used in the literature in the Appendix E. We have now included a reference to Appendix E near that theorem.
>
> **Question 8**
> The authors only present results for the multi-sample upper bounds in the main text. Hence, I don't think presenting the univariate bounds is useful, and their discussion should be moved to Appendix.
>
> **Answer 8**
> We thank the reviewer for providing suggestions on putting the univariate bounds into Appendix. This suggestion is addressed by **AR0.3** above.
> To summarize: As the reviewer pointed out, we put the relevant experiments and results in Appendix H. However, univariate bounds are still useful in general, because when the computational budgets do not allow large batch size in training, the univariate bound would be preferred in training. We have clarified this issue in the last paragraph of Section 4.3.
>
> **Question 9**
> Probably every section title in Section 5 should be reworded. In particular, "Numerical demonstration of the superiority of CPIC" should be renamed to "Results" or "Experiments".
>
> **Answer 9**
> We thank the reviewer for the suggestions. Per the suggestion, we have reworded the subsections in Section 5.
>
> **Question 10**
> "The motivation for using linear forecasting models is that good representations contribute to disentangling complex data in a linearly accessible way" - what does "linearly accessible way" mean?
>
> **Answer 10**
> We thank the reviewer for pointing out the lack of clarity on this statement. We believe this issue is addressed above in **AR0.4**.
>
> **Question 11**
> Just below Eq 3: "$U \in R^{N \times D}$" - what is $U$?
>
> **Answer 11**
> We thank the reviewer for pointing out the typo.  As the notation was not needed, we removed it in the revised version.
>
> **Question 12**
> What is $f_{1}$ and $\gamma$ in Eq 8? In the sentence below, what is $\beta$?
>
> **Answer 12**
> We thank the reviewer for pointing out the typo. We have corrected the notations in the revised version.
>
> **Question 13**
> Label font sizes in Figure 2 and the legend font sizes in Figure 3 should be increased as they are currently hard to read.
>
> **Answer 13**
> We thank the reviewer for pointing out the difficulty in reading the small fonts. We have addressed it in the revised manuscript.
>
> **Question 14**
> Instead of showing mean and best performance, Figure 3 should show median performance with the inter-quartile range as the error bars to give a better idea of performance.
>
> **Answer 14**
> We thank the reviewer providing the suggestion. We have addressed it and added the quantile analysis in the Appendix I in the revised manuscript; conclusions do not change from these differences in summary statistics.
>
> **Question15**
> "Finally, as the SNR gets lower (SNR = 0.001, bottom-right) all methods perform poorly, but we note that, numerically, considering predictive information in latent space is much better than that between latent and observation space." - why is stochasticity better numerically? There is no justification given for this claim.
>
> **Answer 15**
> We thank the reviewer for asking for raising the lack of clarity on this point. To clarify however, the statement makes no reference to stochasticity, only about predictive information in the latent space relative to the observation space. We state that considering predictive information in latent space would has better results. That is illustrated in the Figure 2. In particular, 0.464 (Determinsitic CPIC(L)) $>>$ 0.050 (Determinsitic CPIC(O)) and 0.413 (Stochastic CPIC(L)) $>>$ 0.036 (Stochatic CPIC(O)) As we stated in the Figure 2, we refer (L) to the case with PI in latent space and (O) to the case with PI between latent and observation space. Thus, the statement is justified by the numerical results in the figure.

---

### Official Review · Reviewer_U5Xb · 2022-10-29

**Confidence:** 4
**Correctness:** 3
**Technical Novelty And Significance:** 3
**Empirical Novelty And Significance:** 4
**Recommendation:** 8

**Clarity, Quality, Novelty And Reproducibility:**

It might be more clear to note the differing time-blocks at $T_0$ and $T_1$, since there is no reliance on any symmetry pattern (i.e., there is nothing special about -T versus T as far as I understand.

It might be also helpful to be more clear about the lifting process/simulation for the attractor datasets. While the general idea is conveyed, the exact details it seems could be easily shared (or code given to reproduce the test cases). Overall however this paper seems reproducible.

Overall the bounds are restatements of other results; this is somewhat clear in the paper, but it should still be noted re:Novelty for the purposes of review. There are several other differentiable mutual information estimators, including some that avoid the Gaussian encoder function. Though the paper is already quite thorough, it may be helpful to also test these functions, e.g. the correction of MINE in Song et al 2019 (called SMILE), and the Echo noise encoders in Brekelmans et al 2018.

**Strength And Weaknesses:**

Strengths:
* The method is grounded in established literature (information bottleneck and variational information bottleneck, and more generally Rate-Distortion theory), yet contributes a novel criterion. Secondly, the recognition that the stationarity assumption reduces the learning criterion to a two term loss function, which is, in structure, very similar to many other information trade-offs in the literature.
* Multiple bounds are explored for the mutual information minimization/maximization.
* Experimental results are generally well done, modulo exact generation details.

Weaknesses:
* Allowing time-windows T, for periodic or pseudoperiodic phenomena with period T' we might observe aliasing artifacts. While this remains stationary in a global sense, will such artifacts impede dynamics. Moreover, will the interpretation or downstream use of those dynamics be impeded by this induced false beat frequency at (T-T')/2?
* The variational lower bound (VLB) of Theorem 3 does not appear to require a decoder, contrary to the comment in the sentence immediately following the statement of the theorem; it seems as though it instead requires a good estimate of the conditional likelihood (a prediction from either T to T' or vice versa).


**Summary Of The Paper:**

The authors propose a learning criterion and numerous bounds for learning representations of dynamic systems that are at once maximally compressed (minimal mutual information with their original representation, i.e., minimal rate) while being maximally informative a future time point (or, due to symmetry, a previous timepoint). This leads to learned encodings which are, up to the SNR and trade-off parameter $\beta$,  reflective of the dynamical system.

**Summary Of The Review:**

The authors present an elegant embedding method for dynamical systems, and the estimation machinery to fit the embedding efficiently. The method is novel, but also fits well into existing literature, adding new and interesting directions to Rate-Distortion based encodings. I think the ICLR community at large would be interested in such a manuscript.

---

> ### Author Response · Authors · 2022-11-16
> **Response to Reviewer U5Xb**
>
> Thanks for supporting our paper and providing valuable comments. We've modified our paper based on your comments. Your concerns are addressed as follows.
>
> **Question 1**
> Allowing time-windows $T$, for periodic or pseudoperiodic phenomena with period $T'$ we might observe aliasing artifacts. While this remains stationary in a global sense, will such artifacts impede dynamics. Moreover, will the interpretation or downstream use of those dynamics be impeded by this induced false beat frequency at ($T$-$T'$)/2?
>
> **Answer 1**
> We thank the reviewer for raising the issue of aliasing relevant to the time-windows T. The aliasing artifacts may affect the downstream use of those dynamics. We have provided comments on in Appendix A. Basically, to alleviate the aliasing effect, we need to select the window size carefully by cross validation. Furthermore, we can make plots of the predictive information as a function of both window size $T$ and the embedding dimension $Q$ as diagnostic tools.
>
> **Question 2**
> The variational lower bound (VLB) of Theorem 3 does not appear to require a decoder, contrary to the comment in the sentence immediately following the statement of the theorem; it seems as though it instead requires a good estimate of the conditional likelihood (a prediction from either T to T' or vice versa).
>
> **Answer 2**
> We thank the reviewer for raising the question why Theorem 3 needs a decoder. Directly estimating the mutual information is hard. Similar to (Alemi et al., 2016), we introduced a tractable decoder to get good estimates of the conditional likelihood to derive the VLB. Note that in the VLB, the conditional likelihood is calculated explicitly, and the VLB can be maximized via stochastic gradient descend.
>
> **Question 3**
> It might be clearer to note the differing time-blocks at $T_{0}$ and $T_{1}$, since there is no reliance on any symmetry pattern (i.e., there is nothing special about -T versus T as far as I understand).
>
> **Answer 3**
> We thank the reviewer pointing out the clarity of time-blocks. Therefore, we added more explanation about it such that "Note that $-T$ and $T$ indexes to past and future $T$ data" in that last paragraph of Section 3.
>
> **Question 4**
> It might be also helpful to be clearer about the lifting process/simulation for the attractor datasets. While the general idea is conveyed, the exact details it seems could be easily shared (or code given to reproduce the test cases). Overall, however this paper seems reproducible.
>
> **Answer 4**
> We thank the review asking for the clarity on the lifting process/simulation. This issue is addressed above in **AR0.1**.
>
> **Question 5**
> Overall, the bounds are restatements of other results; this is somewhat clear in the paper, but it should still be noted re:Novelty for the purposes of review. There are several other differentiable mutual information estimators, including some that avoid the Gaussian encoder function. Though the paper is already quite thorough, it may be helpful to also test these functions, e.g., the correction of MINE in Song et al 2019 (called SMILE), and the Echo noise encoders in Brekelmans et al 2018.
>
> **Answer 5**
> We thank the reviewer for providing other references for differentiable mutual information estimators, which we have incorporated into the text (see last second paragraph in Section 4.2, Page 6). The goal of this work is to develop a novel information-theoretic self-supervised learning framework for representation learning with CPIC objective. Estimating the differential mutual information is not the key point. Per the reviewer's suggestion, we have directly indicated that the theoretical results are based on prior works (see **AR0.2** above).

---

### Author Response · Authors · 2022-11-16
**Overview of Reviews and Reply to Common Issues**

We thank the reviewers for their time and helpful comments on our manuscript. We were pleased that many of reviewers found CPIC to be a novel approach for self-supervised learning in the context of high-dimensional time series data. Indeed, most reviewers indicated that the paper should be accepted. Nonetheless, there were several issues that were raised by the reviewers. We reply to all reviewer comments below, and have modified our manuscript where appropriate to incorporate suggestions and address issues. Below, we first address issues that were common across reviewers. We then provide an item-by-item response to each reviewers comments.


**R0.1:** All reviewers pointed out that the details of numerical experiments, analysis, methods are unclear/unavailable, and hence not reproducible.

**AR0.1:** We thank the reviewers for pointing out the lack of clarity on these points, and the negative impact on reproducability. Per the reviewers request, we have provided additional details of simulation in the Appendix G. Furthermore, all software for this paper will be made available upon publication, but is not done at this stage to maintain the integrity of the double-blind review process.


**R0.2:** Several reviewers pointed out that we should clarify that the Theorems are extensions of existing results.

**AR0.2:** We thank the reviewers for pointing out the lack of clarity on this point. We have now clarified this issue in our summary of the primary contributions on Page 2. We note that the goal of this work is to develop a novel information-theoretic self-supervised learning framework for representation learning with CPIC objective. Estimating the differential mutual information is not the key point.
Specifically, we state:
"Based on prior works, we derived varitional bounds....."


**R0.3:** Several reviewers suggest that since only results for multi-sample bounds are used in numerical experiments, it would be better to spend less time on the unisample results in the main text.

**AR0.3:** We thank the reviewers for this suggestion to stream-line the presentation of the results. However, we note that univeriate bounds are still useful in general, because when the computational budgets do not allow large batch size in training, univeriate bounds would be preferred in training. We have clarified this issue in Section 4.3 (middle of Page 6):". First, the performance of multi-
sample upper bound depend on batch size while uni-sample upper bounds do not, so when computational budgets do not allow large batch size in training, uni-sample upper bounds may be preferred
in training. Secondly, multi-sample upper bound has lower variance than uni-sample upper bounds. Thus, they have different strengths and weaknesses depending on the context."


**R0.4:** Several reviewers requested more motivation as to why we elected to use only linear models for prediction of external variables, and explain why metrics are modest for all cases.

**AR0.4:** We thank the reviewers for pointing out the lack of clarity on these points. The motivation for using linear models to predict exogenous variables is that end users, e.g., neuroscientists, often want to interpret latent representations of data to gain insight into the processes that generate the observed data. The intuition is that a simple (i.e., linear) prediction model will only be sensitive to the structure in the data that is easiest to interpret (i.e., the structure that is linearly related to the prediction task). Thus, we used a linear model of decoding. With regard to the magnitude of the evaluation results on real data, we note that the neuroscience data sets (M1 and HC) present extremely challenging settings for prediction of the exogenous variables due to severe experimental undersampling of neurons due to technical limitations, as well as sizeable noise magnitudes. Nonetheless, the numerical results clearly demonstrate that superiority of the proposed method relative to several competing methods. More broadly speaking, we believe that utilizing real-world scientific data sets is an important direction for application of machine learning methods. Neuroscience data sets in particular have been and continue to be important motivators for novel methods development. We have clarified these points and commented on them in section 5.2 (Page 9): "Neuroscientists often want to interpret latent representations of data to gain insight into the processes that generate the observed data. Thus, we used linear regression to predict exogenous variables, with the intuition that a simple (i.e., linear) prediction model will only be sensitive to the structure in the data that is easiest to interpret. Furthermore, the neuroscience data sets (M1 and HC) present extremely challenging settings for prediction of the exogenous variables due to severe experimental undersampling of neurons due to technical limitations, as well as sizeable noise magnitudes."

---

### Decision · Program_Chairs · 2023-01-20

**Decision:**

Reject

**Justification For Why Not Higher Score:**

The experimental section is not reproducible and the comparison to other methods might not be fair.

**Justification For Why Not Lower Score:**

N/A

**Metareview: Summary, Strengths And Weaknesses:**

In this paper, the authors propose a new information bottleneck criteria to find a latent representation for time series. They provide a novel self-supervise learning criterion trained using a variational lower bound (See the summaries by the reviewers for more information).

The strongest results in the paper are in Section 3, where the authors present the algorithm. All the reviewers liked this section, especially reviewer U5Xb is enthusiastic about this result. The algorithm is compelling and belongs at a major ML conference.

The weakest aspects of the paper are the misleading Section 4 and, especially, the experimental results in Section 5. The reviewers point out that the theorems in Section 4 are the application of existing results to the proposed problem. Section 4 should be an easy fix.

The main issue with the paper is the experimental results that are not reproducible. The authors have tried to respond to these criticisms, but the needed work is more fundamental. For example, in the Lorenz attractor results, the authors simulate the 3D time series and linearly map it to a 30-dimensional space (random matrix). They use their procedure to map it back to a 3D space. From Figure 2, top row, it seems that they recover exactly the Lorenz attractor trajectory. How they can avoid rotations or permutations is not explained. If this is not an issue, it would be fundamental to know why.

The experiments with real data are also not well-detailed and, it is impossible to know if the comparisons are fair to the other methods. To validate the results, they run a prediction task in which the authors mention that they use a linear classifier for interpretability, but if the encoder is nonlinear, the features found by the proposed procedure are not interpretable anyway. Also, we do not know the values for Q in these experiments or the structure of the neural nets used for each time series.

Finally, the authors mention that T is either 1, 2, or 3. But they do not say which one it is. Even for T=3, the encoder does not look into a significant portion of the time series. T = 3 is too short for that. But for T=1 the encoder only looks at independent samples. It would be very important to understand the role of T in the algorithm and how it does process the data. If T=1 provides the best solution, the role of the encoder would be very different for a T>1 or even for a T that is in the tens of samples. This aspect needs to be studied before the paper can be published.

I apologize to the authors for bringing new criticism to the experimental section of the paper. These issues arouse in a zoom meeting with the reviewers to decide if we should accept the paper for ICLR. We encourage the reviewers to improve the paper and submit it next year.

**Summary Of Ac-Reviewer Meeting:**

In the review meeting, reviewers U5Xb (most positive reviewer) and aQPA (neutral) were at the meeting. Reviewer 3inW could not make it, but I have exchanged emails with him about the meeting. jyD8 did not respond to any of my emails.

The metareview above explains well what the meeting was about. U5Xb mentioned his liking for Information Theory based papers, but he understood the limitation of the experimental section. He agrees that it needs significant work before it can be published. Reviewer aQPA and 3inW also agreed to reject it.